# A Transmissive Imaging Spectrometer for Ground-Based Oxygen A-Band Radiance Observation

Heng Wu [1,2], Junqing Wu [3], Nanxi Hu [3], Hang Cui [3], Pengfei Wu [4], Guanyu Lin [1], Diansheng Cao [1,*], Zihui Zhang [1], Yingqiu Shao [1] and Bo Li [1]

1   Changchun Institute of Optics, Fine Mechanics and Physics, Chinese Academy of Sciences, Changchun 130033, China
2   University of Chinese Academy of Sciences, Beijing 100049, China
3   Beijing Institute of Control and Electronic Technology, Beijing 100038, China
4   Key Laboratory of Atmospheric Optics, Anhui Institute of Optics and Fine Mechanics, Chinese Academy of Sciences, Hefei 230031, China
*   Correspondence: caodiansheng1987@163.com

**Abstract:** The oxygen A-band (759–770 nm) is a commonly used band for atmospheric observations. The signal in this band has wide dynamic range and can be used to invert several atmospheric parameters, such as air pressure and atmospheric optical depth, at different altitudes. High-resolution oxygen A-band radiance imaging spectrometer (HARIS) is an imaging spectrometer that operates in the oxygen A-band, which is designed for the observation of the direct solar radiance that passes through the atmosphere. HARIS is a transmissive imaging spectrometer that uses a compact transmissive optical system combined with reflective grating spectroscopy, while an area scan CMOS detector is used as the photosensitive element for the observations. HARIS response is associated with the observed target through a calibration process, which uses a monochromator with a supercontinuum laser for the spectral calibration, an integrating sphere with a spectrophotometer for the radiometric calibration and a meridian for the geometric calibration is employed to correct for distortions. The calibration results show that HARIS has an average spectral resolution of 0.33 nm and a field-of-view of $3.085 \times 0.03°$ with an average spatial sampling interval of 0.0138°. Finally, the performance of HARIS is verified through field tests, in which the solar radiance data with an average signal-to-noise ratio of 438.93 is obtained.

**Keywords:** oxygen A-band; transmissive imaging spectrometer; spectrum calibration; absorption spectrum

## 1. Introduction

The oxygen A-band is one of the most commonly used spectral bands in ground-based and space-borne atmospheric remote sensing, with a spectral range of 759–770 nm. The atmospheric absorption spectral line distribution on the oxygen A-band is regular, the signal dynamic range is large, and the absorption intensity varies drastically. The above characteristics make it an ideal band for atmospheric parameter inversion [1–7]. Cloud top-height inversion, atmospheric optical depth inversion, and daylight-induced chlorophyll fluorescence analysis can be performed by analyzing the intensity of the radiation signal [8–12]. Atmospheric pressure profile inversion can be performed by analyzing the spectral signal broadening of the absorption band [13]. The radiation transmission patterns can be improved, and the cloud 3D effects can have a greater understanding, by analyzing the photon transport paths in the absorption band [14]. The aerosol profile information can also be inverted by use of the oxygen A-band polarization signal [15]. Furthermore, on-orbit upper atmospheric oxygen A-band airglow emission observations can be used for the inversion of atomic oxygen densities [16].

With the growth of greenhouse gas detection requirements and the improvements in precise spectroscopic measurement techniques, remote sensing technology (and its applications with respect to the oxygen A-band) is beginning to experience a new period of rapid development. A number of advanced spectrometers are now employed for oxygen A-band observations. Among these, the greenhouse gases observing satellite-2 (GOSAT-2) is able to use its Fourier transform spectrometer-2 (TANSO-FTS-2) with a biaxial scanning mirror to examine multiple observation areas, which are 8 × 88 km in size over a span of 790 km [17,18]. The TanSat (a carbon observing satellite) is able to observe sub-satellite areas with a ground sample distance of 2 × 2 km over an area of 20 km in width [19]. The Earth Polychromatic Imaging Camera (EPIC) is able to observe the Earth disk at the L1 point with 8 km spatial resolution at sub-satellite points [20,21].

The above-mentioned instruments are able to obtain remote sensing data for the oxygen A-band from space with global coverage at high spectral resolution and high spatial resolution. However, these space-borne instruments have the weakness of a limited temporal resolution for the remote sensing observations, a relatively fixed visit time over each region, and a satellite attitude-limited observation angle. Therefore, ground-based instruments are also required to obtain a high temporal resolution and a multi-angle data for certain regions [22].

In terms of the ground-based instruments, high-resolution oxygen A-band and water vapor band spectrometer (HAWS) and high-resolution oxygen A-band spectrometer (HABS), which use grating for spectroscopy, have obtained oxygen A-band radiation data with high resolution. The HABS has an excellent spectral resolution (i.e., 0.016 nm) and it has a frontal telescope with a 2.71° field-of-view (FOV) that can be adjusted for observing both direct solar radiation and the sky scattering signals. However, the large structure of the instrument and the accompanying cooling equipment means that it is inconvenient to deploy in the field [7,23]. Although rotating shadowband spectrometers (RSS) are also capable of obtaining radiation data in the oxygen A-band, their spectral resolution is low (i.e., 2.3 nm) [24]. In recent years, researchers have also designed ground-based hyperspectral instruments for oxygen A-band observations, such as the double-grating spectrometer system (DGSS) and simultaneous multipolarization and high-resolution oxygen A-band spectrometer (SPHABS), but they have not yet been in use or have large dimensions [25–27].

Simultaneously, for the application of oxygen A-band remote sensing data, if the data has a high spectral resolution, signal-to-noise ratio (SNR), out-of-band rejection ratio (OOB), and suitable spectral sampling interval, it can provide more independent information for the inversion of atmospheric parameters [23,28]. In addition, direct observations of direct solar radiation can also provide more atmospheric information, such as photon path length distributions, and higher SNR [7]. Therefore, it is necessary to use the hyperspectral instruments with high spatial resolution, high spectral resolution and high SNR to observe the atmosphere in the oxygen A-band. However, compact and easily deployable observing instruments that are designed for this purpose are rarely reported in the literature.

In this paper, a compact imaging spectrometer for the oxygen A-band observation, the high-resolution oxygen A-band radiance imaging spectrometer (HARIS), is proposed and designed, which is able to observe direct solar radiation with a high spectral resolution and a high SNR under uncooled conditions. The design and the calibration of the instrument are introduced, and this is followed by the description of the field test and a brief analysis on the field test results. These outcomes show that the instrument can meet observation requirements for oxygen A-band radiance with a high spatial resolution and a high SNR.

## 2. Instrument Design and Calibration

### 2.1. Optical Design

To reduce the size and the weight of the optical system, HARIS has been designed as a transmissive imaging spectrometer. All the components of HARIS except for the grating are transmissive, and the layout of the optical system is shown in Figure 1. The incident light radiation is imaged onto the focal plane after it sequentially passes through

the band-pass filter, neutral density filter, telescope system, slit, collimation system, grating, and focusing system.

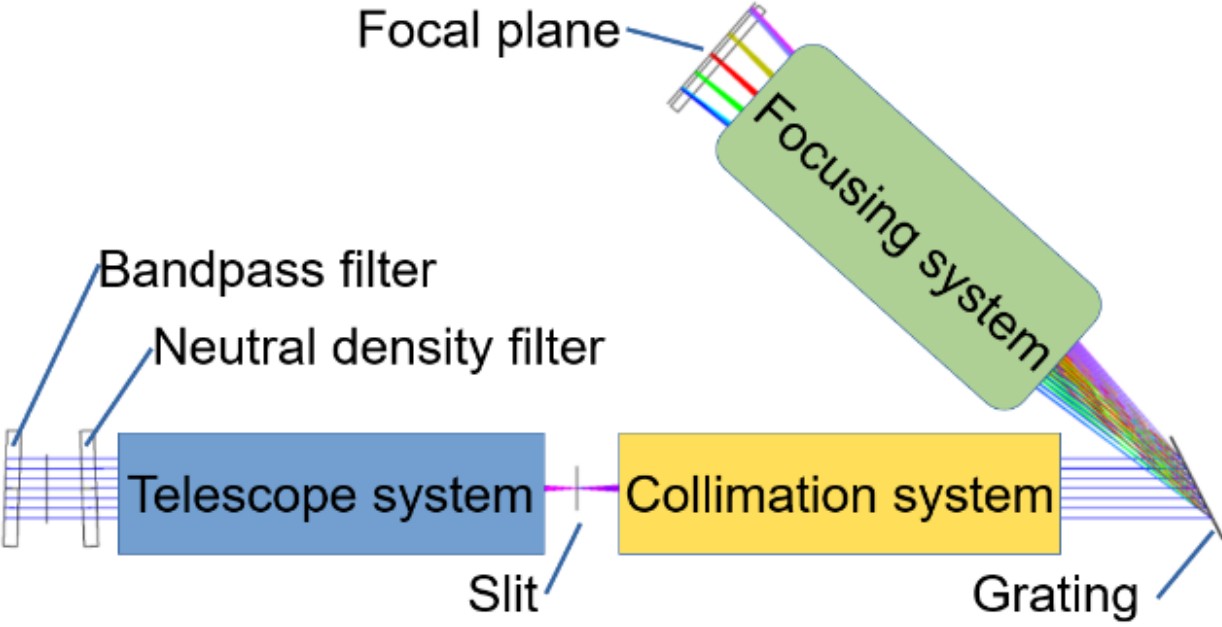

**Figure 1.** Layout for the optical system based on HARIS.

The bandpass filter in the system serves to eliminate stray light and the higher-order spectral effects from the non-observation bands, which also improves the OOB levels of the instrument; the neutral density filter is used to attenuate the excessive sunlight to a level acceptable for the detector. The focal length of both the telescope and the whole optical system of HARIS is 55 mm. The slit is positioned at the focal point of the telescope system and has a width of 25 μm, which limits the FOV of the spectral dimensions of the instrument to 0.03°. A holographic planar grating with 1200 lines/mm is used. The detector resolution is 2048 × 2048 and the pixel size is 6.5 × 6.5 μm. The designed operating band covers from 758 nm to 778 nm, the designed spectral resolution is greater than 0.5 nm, and the designed FOV is 3 × 0.03°.

*2.2. Image Output Method*

To achieve high SNR measurements for uncooled conditions, HARIS uses a method of pixel binning with the simultaneous recording of reference dark column signals to reduce the effects of noise [29,30]. From the simulation results, the image plane of the optical system occupies approximately 2000 rows and 510 columns of the CMOS detector. To suppress the effect of noise and stray light as much as possible for the uncooled case, HARIS also records 32 columns of unilluminated pixels that are a distance of 300 columns away from the effective detector region. This data will be used as the reference dark signal of the unilluminated condition, which is used for the spectral information extraction. A total of 2040 rows and 518 columns of the images that contain effective spectral information is recorded for each observation, which leaves a slight margin for the effective region. So, a total of 2040 rows and 550 columns of the image data need to be recorded for the same observation.

Due to compactness requirements, HARIS is not able to improve the SNR of the data through detector cooling components. Therefore, the SNR of the observed data is planned to be improved by pixel binning. The noise of the signal acquired by the CMOS detector mainly consists of thermal noise, which obeys the statistical law, and the readout noise brought by the circuit design. The pixel binning assists in reducing the influence of the thermal noise of the signal, which results in the improvement of the SNR for the observed data.

From the optical design results, the detector has a redundancy in both the spatial and the spectral dimension sampling intervals for the observed signal. Thus, HARIS is designed to output the sum of every 10 rows and every 2 columns of the original detector data, so that the 20 pixels are combined into 1 pixel, which provides a resolution of 204 × 275 for the final output data. The image transmission process of HARIS is shown in the form of a schematic in Figure 2.

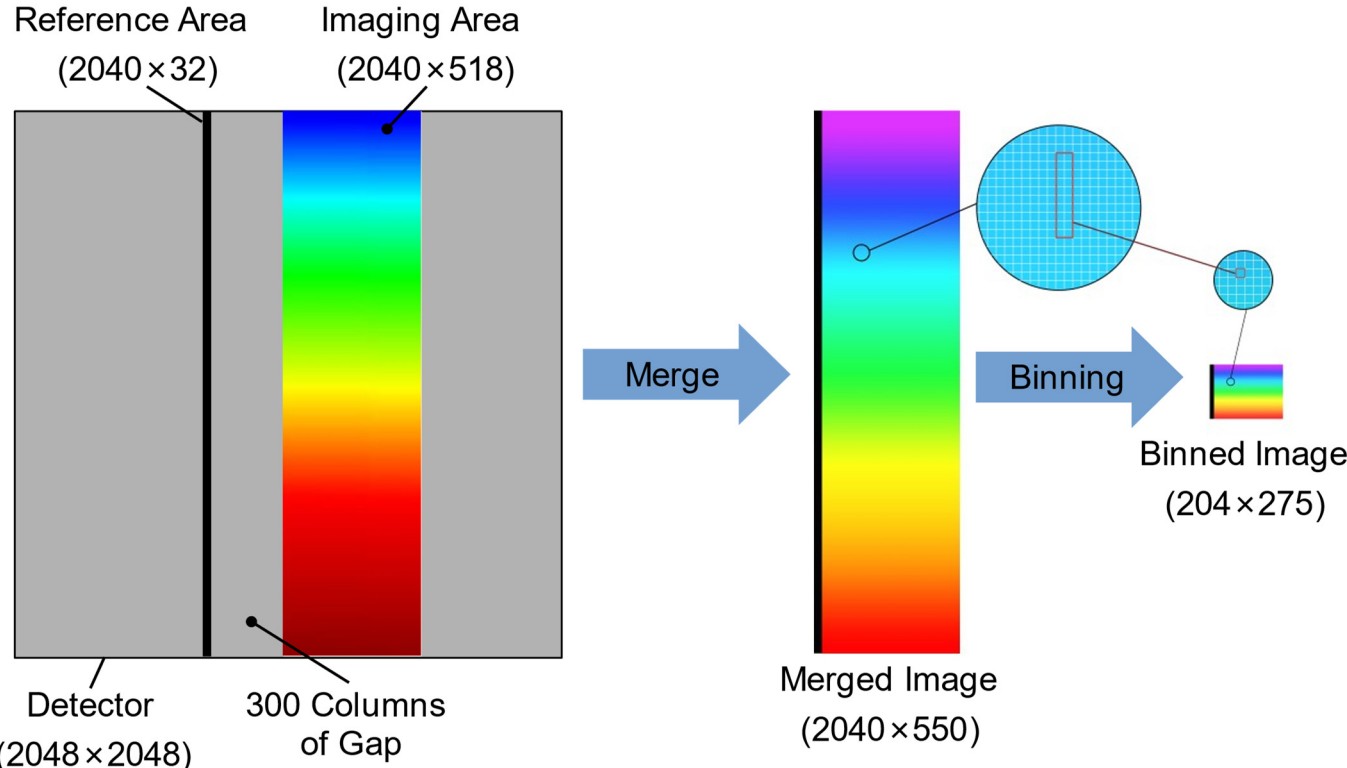

**Figure 2.** Image transmission process for HARIS.

In the case of 20-pixel binning, after excluding pixel readout noise and bias (because these two values are generally constant and can be eliminated by calibration), the digital number after pixel binning $DN_{binning}$ is the sum of the digital number of each pixel before binning. During the pixel binning, the digital number from the optical signal conversion $L_i$ are directly summed linearly and the random noise $N_i$ is summed statistically. Thus, the result of pixel merging can be calculated by the following equation

$$DN_{binning} = \sum_{i=1}^{20} L_i + \sqrt{\sum_{i=1}^{20} N_i^2} \, , \tag{1}$$

The sum of the random noise can also be approximated by the following equation

$$\sqrt{\sum_{i=1}^{20} N_i^2} \approx \sqrt{20} \, \overline{N_i} \, , \tag{2}$$

where $\overline{N_i}$ is the average of all pixel random noise. Then, if we allow the average of all pixel response signal to be $\overline{L_i}$, the combined SNR can be calculated as

$$SNR_{binning} = \frac{20\overline{L_i}}{\sqrt{20}\overline{N_i}} \, , \tag{3}$$

Meanwhile, the average SNR before merging can be expressed as $\overline{N_i}/\overline{L_i}$, so after 20 pixels of merging, the SNR of the data is improved by a factor of $\sqrt{20}$.

### 2.3. Wavelength Calibration

The accuracy of the wavelength calibration directly affects the accuracy of the inversion results using the observational data [31]. The calibration method using the multipoint monochromatic light is only able to fit the central wavelength on the image plane, without the acquisition of the full width at half maxima (FWHM) of the slit function simultaneously. Therefore, HARIS is calibrated pixel-by-pixel to obtain the slit function at each position of the image plane. The equipment required for the wavelength calibration of HARIS is shown in Figure 3. which consists of an NKT EXB-6 supercontinuum laser, a McPherson Model 209 monochromator, and supporting optical components. After a beam expansion and homogenization, the light emitted from the supercontinuum spectrum laser enters the entrance slit of the monochromator. Then the beam comes out of the exit slit and, after a color separation by the monochromator, is then expanded and imported into HARIS to achieve wavelength calibration.

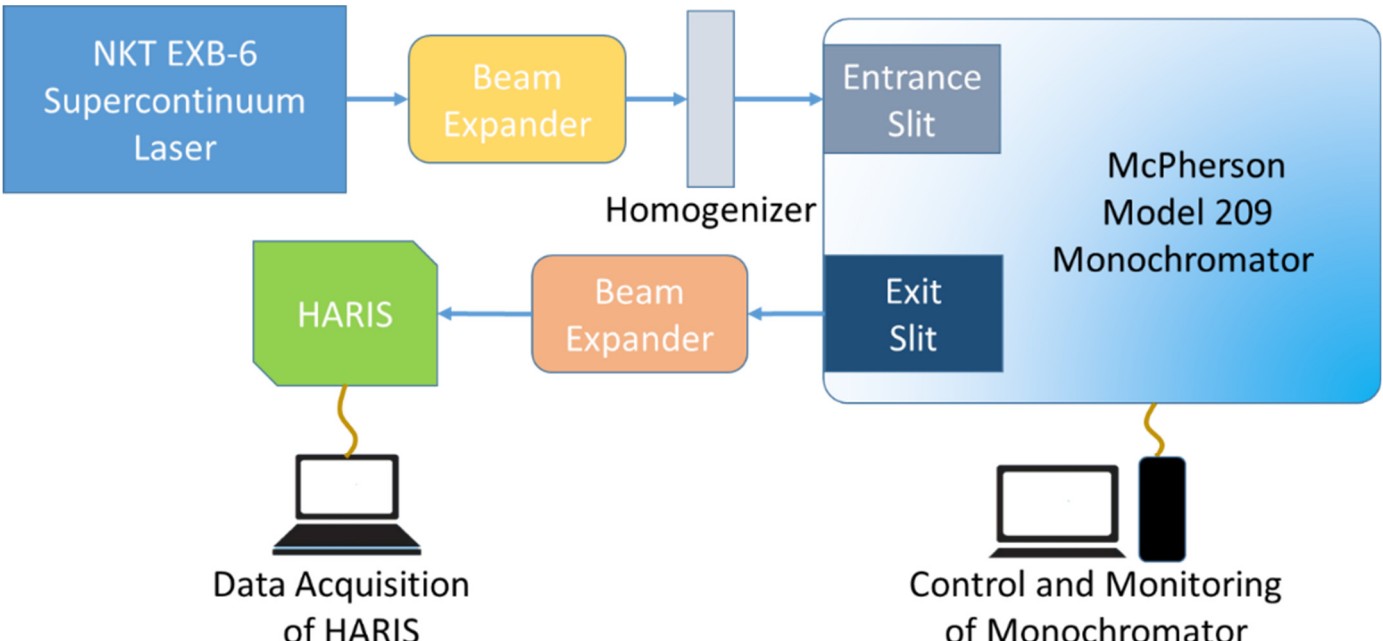

**Figure 3.** Diagram depicting the equipment for the wavelength calibration of HARIS.

After calibrating the monochromator, the wavelength calibration of HARIS was performed using the above system. For this optical design, the wavelength scan interval for the pixel-by-pixel calibration was taken as one-third the amount of the designed spectral resolution at 0.15 nm, and a total of 148 points were scanned from 757 to 779.05 nm. Figure 4 shows the response of the 150th column on the HARIS image plane at 770.05 nm.

After the acquisition of the wavelength calibration data, the response of each pixel to different wavelength light signals were fitted with the Gaussian curve, and the slit function of each pixel on the image plane was obtained. Figure 5 shows the slit function of the 150th pixel in the 120th row. The central wavelengths that correspond to the pixels in the 150th column are presented in Figure 6, which depicts a nonlinear distribution of wavelengths on the image plane.

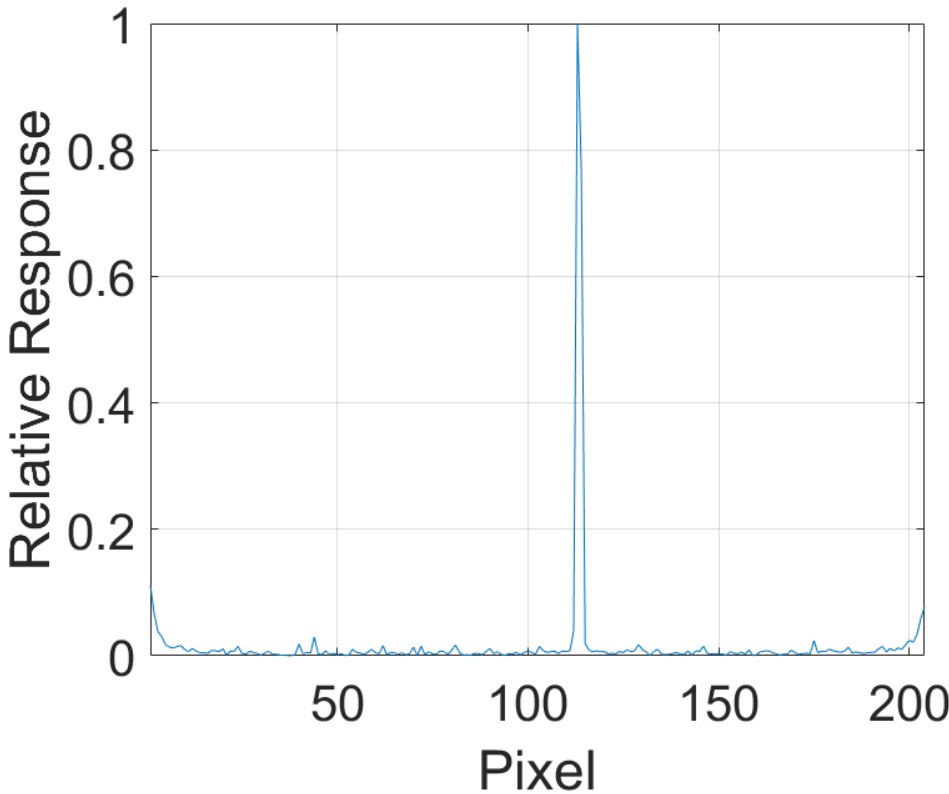

**Figure 4.** Response of the 150th column on the HARIS image plane at 770.05 nm.

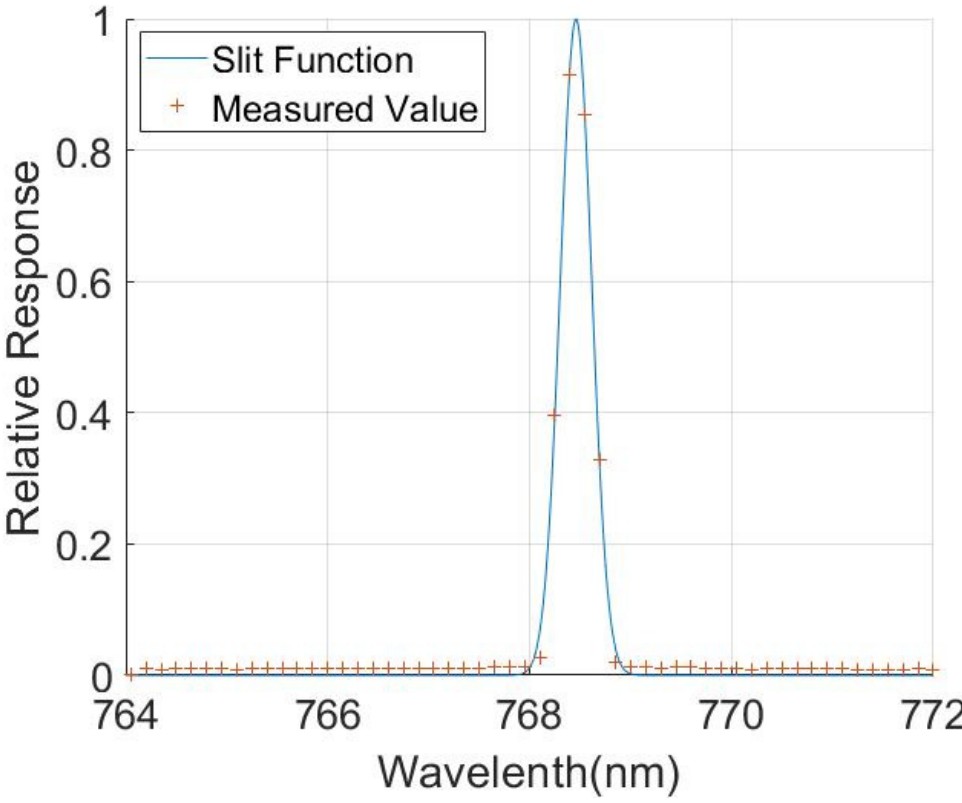

**Figure 5.** Slit function of the 150th pixel in the 120th row, which corresponds to the central wavelength at 768.46 nm and a FWHM of 0.36 nm.

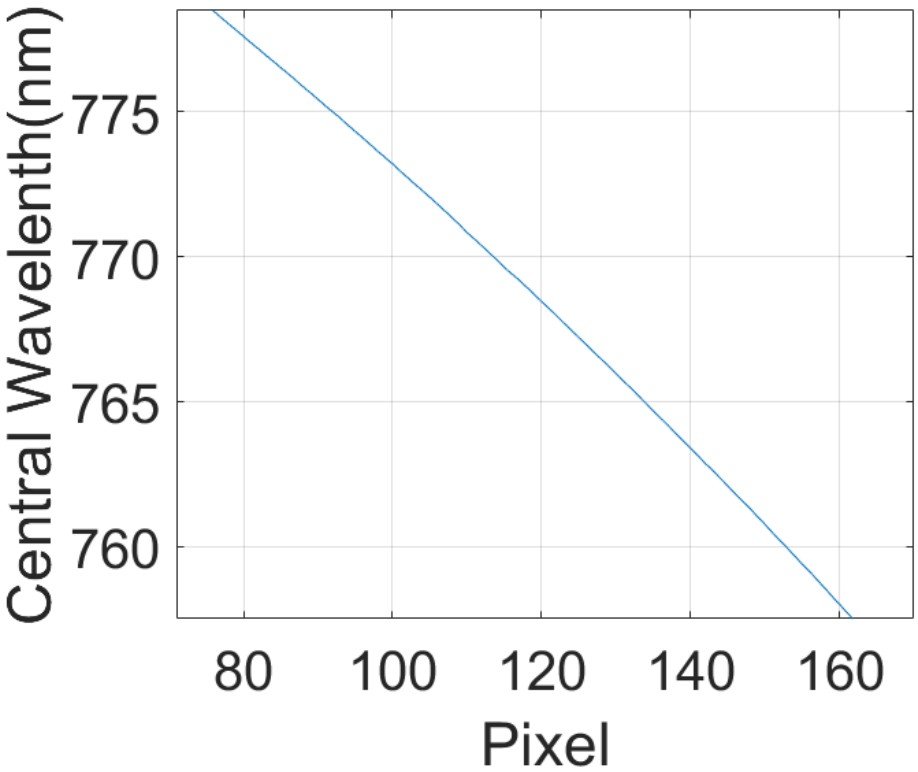

**Figure 6.** Central wavelengths that relate to the pixels in the 150th column.

After the wavelength calibration, it can be determined that HARIS is able to cover the spectral band from 758 nm to 778 nm with an average FWHM of 0.33 nm, and an average spectral sampling interval of 0.25 nm.

### 2.4. Radiometric Calibration

As the target of HARIS observations is the Sun, and the currently used integrating sphere light sources are not able to simulate high brightness targets such as the Sun, direct calibration is difficult. The HARIS system has a neutral density filter with a transmittance that is approximately 1.4%, the removal of which can significantly improve the radiometric responsiveness of the instrument. Therefore, HARIS calibration is separated into two parts: one for HARIS spectral irradiance responsivity calibration without the neutral density filter, and the other for the spectral transmittance measurement of the neutral density filter.

The spectral radiance responsiveness was calibrated using an integrating sphere with a spectral radiance, near to 760 nm, of 0.453 W·m$^{-2}$·nm$^{-1}$·sr$^{-1}$. HARIS spectral irradiance responsivity without the neutral density filter, $R_{without}$ can be calculated based on the standard spectral radiance of the integrating sphere $L_{IS}$ (which was transmitted in advance), the digital number $DN$, and the integration time $t_{int}$ during the calibration, so that:

$$R_{without} = \frac{DN}{L_{IS}t_{int}} ,\qquad(4)$$

The spectral transmittance measurements of the neutral density filters were performed using a spectrophotometer. The spectral transmittance data for the neutral density filter was obtained by averaging the measurements of the filter within the 750–790 nm band. Due to the presence of the neutral density filter, the signal received by HARIS is reduced, i.e., there is a relationship between the spectral irradiance of the observed target $L_{target}$ and

the spectral radiance received by the subsequent optical system of HARIS $L_{opt}$ and the transmittance of the neutral density filter $T$, given as follows:

$$L_{target} = \frac{L_{opt}}{T},$$ (5)

By combining Equations (2) and (3), the spectral radiance responsiveness of HARIS, $R_{HARIS}$, is calculated as:

$$R_{HARIS} = \frac{DN \cdot T}{L_{opt} \cdot t_{int}}.$$ (6)

It is noteworthy, as shown in Figure 7, that HARIS also suffers from the Etalon effect due to the use of a back-illuminated CMOS detector [32]. However, because of the high stiffness of the HARIS optical structure, this effect can be circumvented by ensuring that the image plane position during the observation is essentially the same as it was during the calibration.

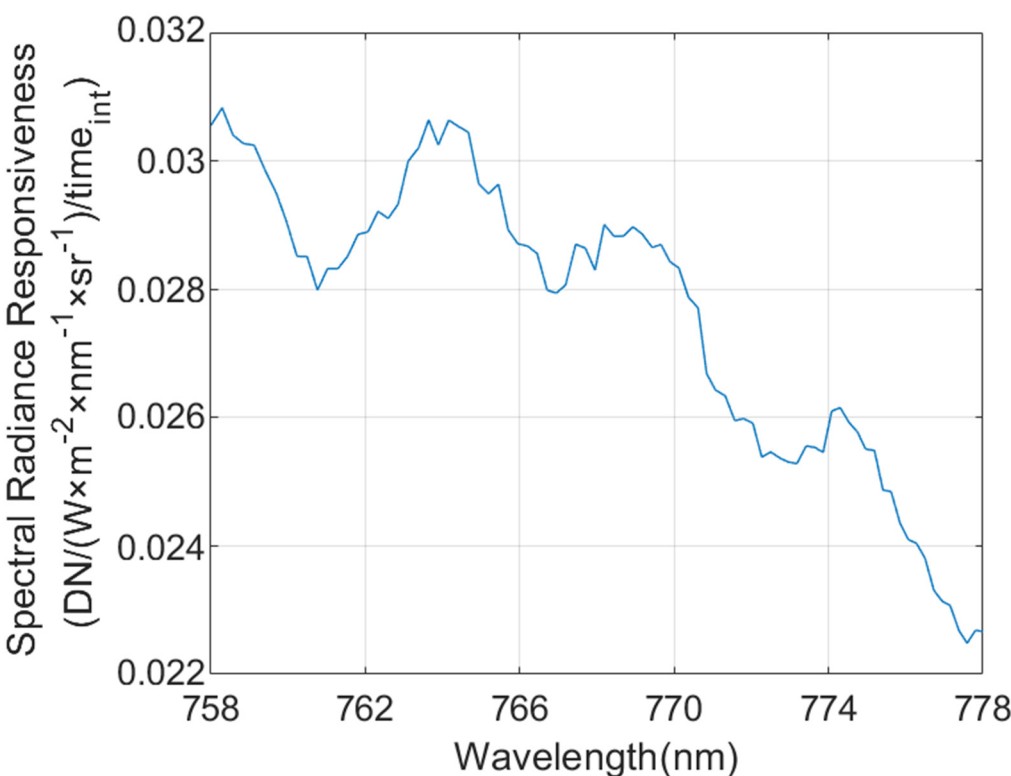

**Figure 7.** Spectral radiance responsiveness of the 115th column.

## 2.5. Geometric Calibration and Correction

Due to the residual distortion within the optical design and the errors in the processing and the mounting, there are some distortions in the imaging results (as shown in Figure 8). In addition, since HARIS is an instrument for observing the sky with a high spatial resolution, it is also necessary to determine the pixel–spatial position relationship. Therefore, a geometric calibration and a correction are required.

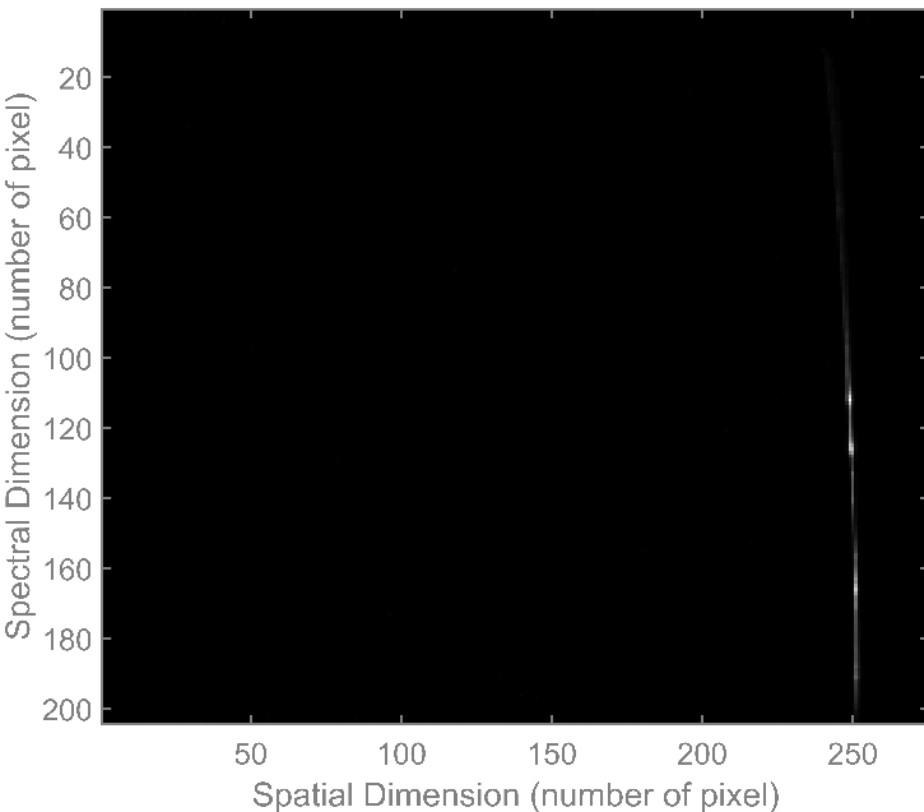

**Figure 8.** Response of HARIS at the geometric calibration of 88.57°. As shown, the instrument response is a curve in the spectral dimension before the calibration is performed.

The equipment for the geometric calibration of HARIS is shown in Figure 9, which includes a xenon lamp, a collimator, a slit placed at the focal point of the collimator, and a theodolite. The slit width $w_o$ is around 100 μm, the focal length $f_c$ of the parallel light tube is 507.16 mm, and the focal length of HARIS, $f_{HARS}$, is 55 mm. The theoretical calculations can be obtained, for a width $w_i$ of the slit image on the image plane, from the following expression:

$$w_i = \frac{w_o \cdot f_{HARS}}{f_o} = 10.84 \text{ μm.} \tag{7}$$

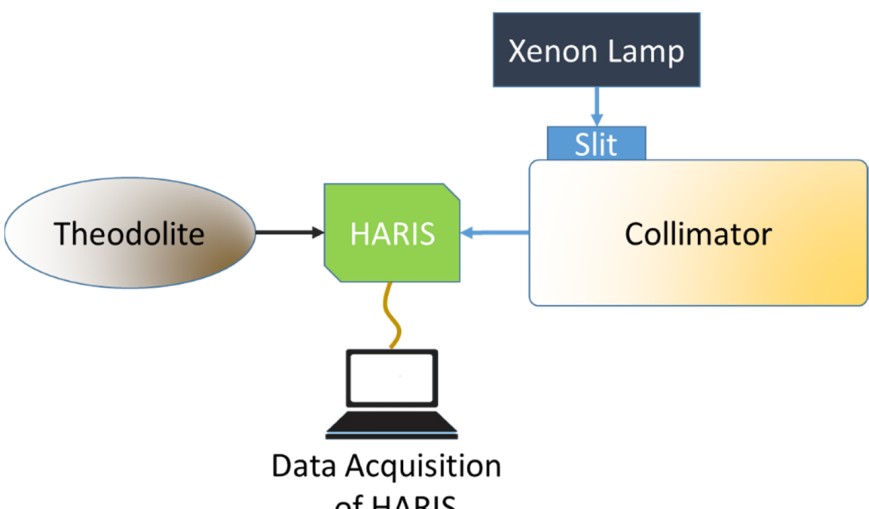

**Figure 9.** Diagram depicting the equipment for the geometric calibration of HARIS.

This is slightly smaller than the width of the binned pixel (i.e., 13 µm). The slit on the collimator is aligned perpendicular to the slit inside HARIS to limit the FOV. The light from the xenon lamp is incident onto the collimator, after it passes through the FOV limiting slit, then incident onto HARIS. With the changing of the tilt angle of HARIS using the adjustment table, the response of HARIS to different incident angles of parallel light can be recorded. While adjusting the tilt angle of HARIS, the tilt angle was monitored and then recorded with the theodolite, and the calibration data were collected at an interval of around 6 arc minutes. Figure 8 shows the response of HARIS when the angle between HARIS optical axis and the vertical direction is 88.57° during the geometric calibration. This shows that there is some distortion at the edge of the image.

After completing the data acquisition of the geometric calibration, each set of data was analyzed and the Gaussian fitting was performed row-by-row to obtain the position of the incident light signal that is received by the image plane at different angles. The response of the 120th row at 89.99° is shown in Figure 10 with its Gaussian fit. The final image position–observation angle correspondence for the full image plane was obtained by use of polynomial fitting results for the image position–observation angle relationship of all the rows. The pixel–space angle corresponding to the 120th row is shown in Figure 11 which shows the angle between each pixel and the instrument optical axis. The geometric calibration results show that the instrument FOV is 3.09 × 0.03°, and the average spatial sampling interval is 0.0138°.

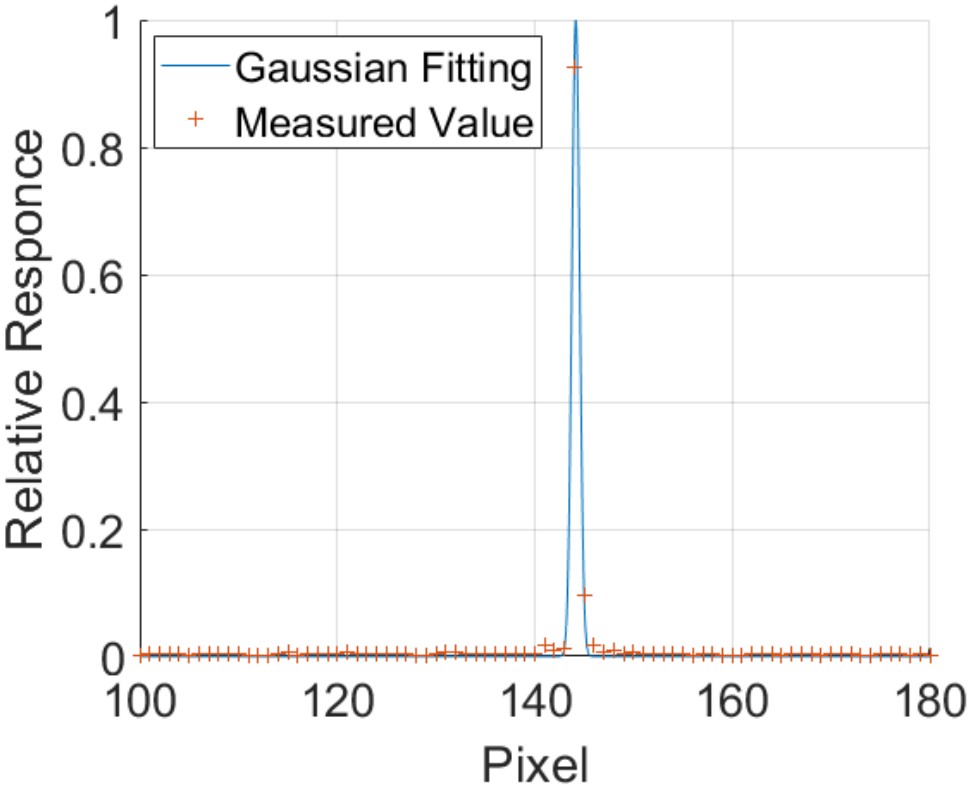

**Figure 10.** Response of the 120th row of HARIS when receiving an 89.99° incident signal and its fitting results, for a central position of 144.15 and a FWHM of 0.92.

Using the geometric calibration results, the distortion corrections for HARIS can be determined. The correction method uses the FOV of the 102nd row as the reference, which is located at the center of the image plane, then interpolates the data of the other rows to obtain the corrected image; the results for which are displayed in Figure 12.

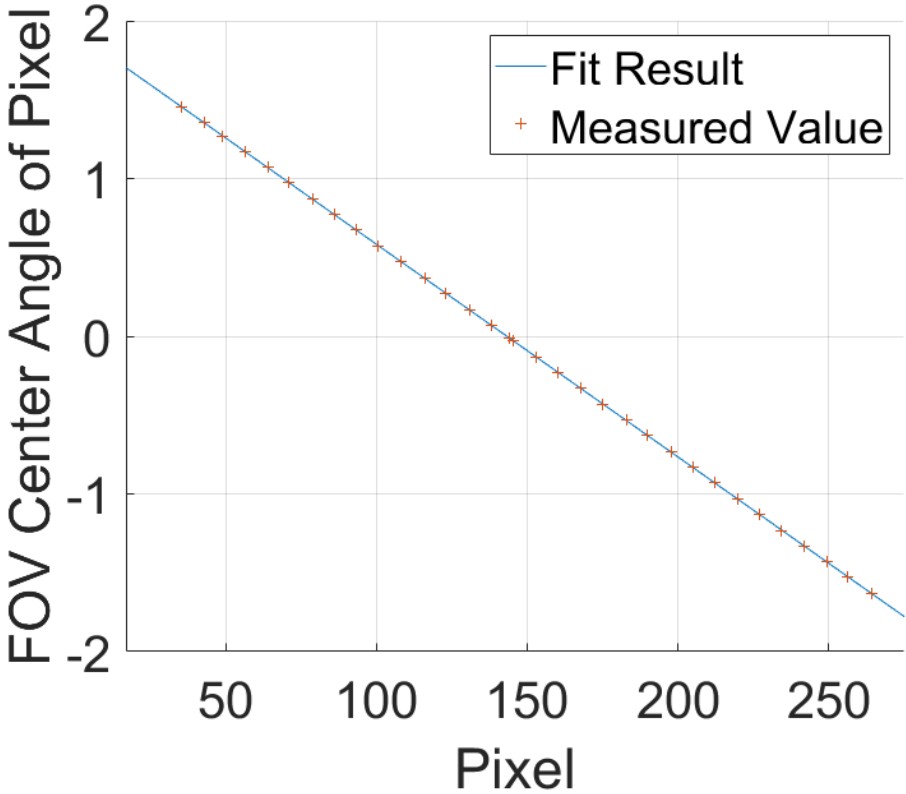

**Figure 11.** Pixel-space angle that corresponds to the 120th row.

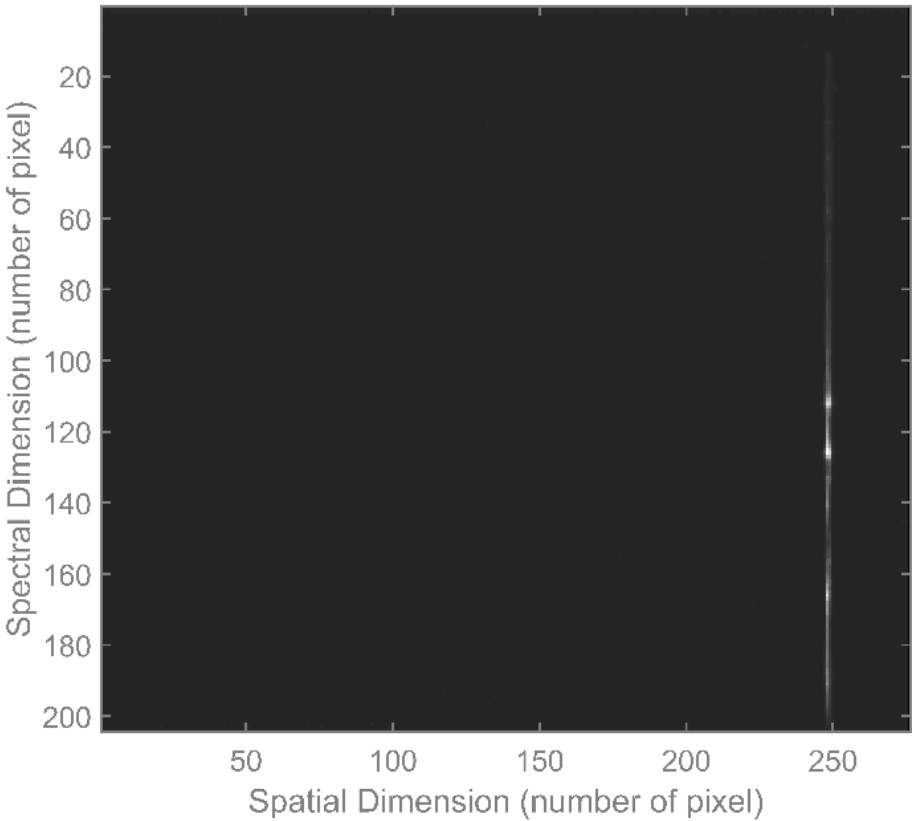

**Figure 12.** Corrected HARIS response at the geometric calibration of 88.57°. After correction, the instrument response becomes a straight line in the spectral dimension.

## 3. Field Test

### 3.1. Overview of Field Test

To verify the performance of the HARIS system, it was tested in October and December 2021 at the Lijiang Observatory of the Yunnan Astronomical Observatory, CAS (latitude 26°41′24.97″ N, longitude 100°01′49.91″ E, and altitude 3200 m) during two field tests. The Lijiang observatory is located on the top of Tiejia Mountain, which is around 40 km from downtown Lijiang; it has the advantage of a high atmospheric visual nimbleness and a low aerosol impact. The setup used for the field tests is shown in Figure 13, in which HARIS is mounted onto the Paramount ME II equatorial mounting for solar tracking. This equatorial mounting is capable of achieving a 30 arcsecond celestial pointing accuracy and a 7 arcsecond tracking accuracy that ensures that the high-resolution observations of HARIS are not affected by the pointing accuracy.

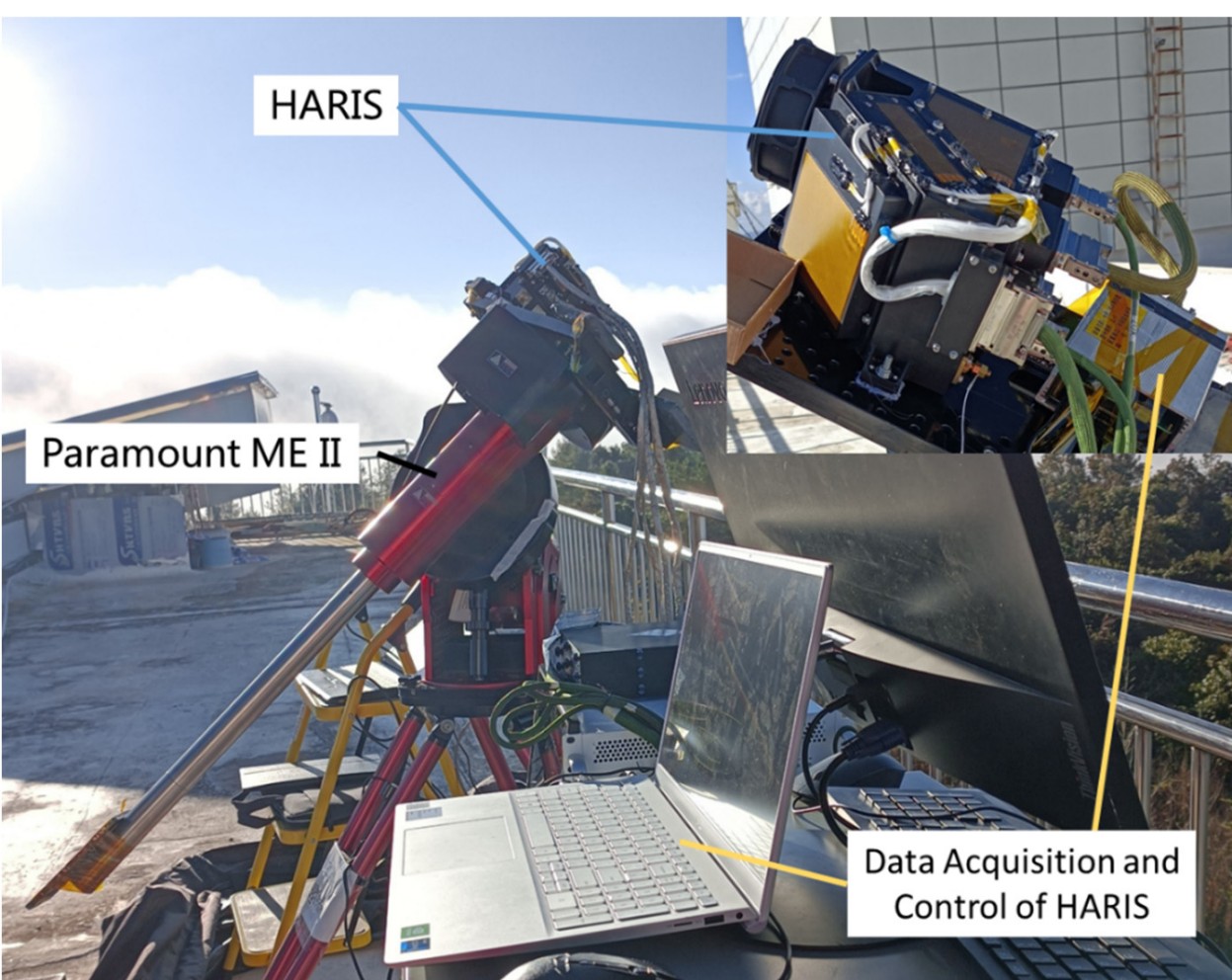

**Figure 13.** Photograph of the setup that was used for the field tests.

### 3.2. Field Test Data and Analysis

During the tests, observations were conducted by HARIS at multiple solar altitude angles. Figure 14 shows the spectral radiance values, and their relative values, during the two observations on 9 October and 1 December at the same solar altitude angle of 41.263°. These two images demonstrate that HARIS is capable of achieving observations of the oxygen A-band, and Figure 14b shows that the observation results of the instrument are reproducible under similar conditions.

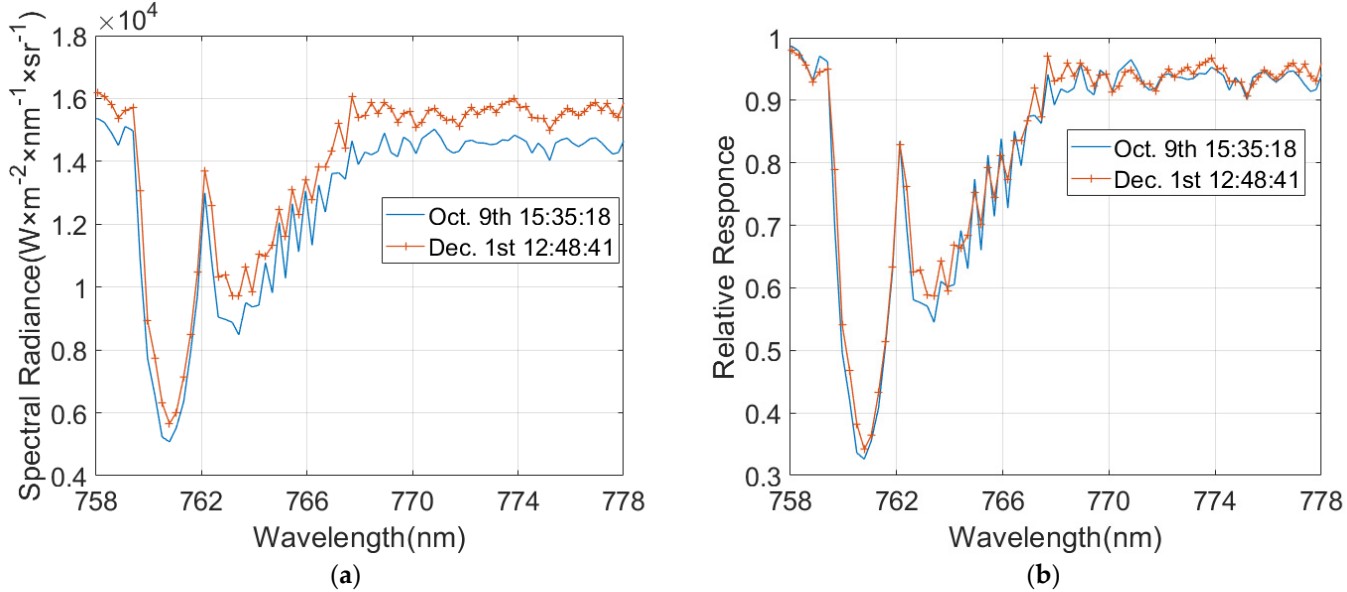

**Figure 14.** Spectral radiance data (**a**) and relative response (**b**) of HARIS obtained from the two observa-tions in October and December.

The atmospheric conditions during these two observations are displayed in Figure 15. There was a cloudy sky on 9 October and a clearer sky on 1 December; this difference is also reflected by the variation in the absorption intensity of the oxygen A-band between the two observations.

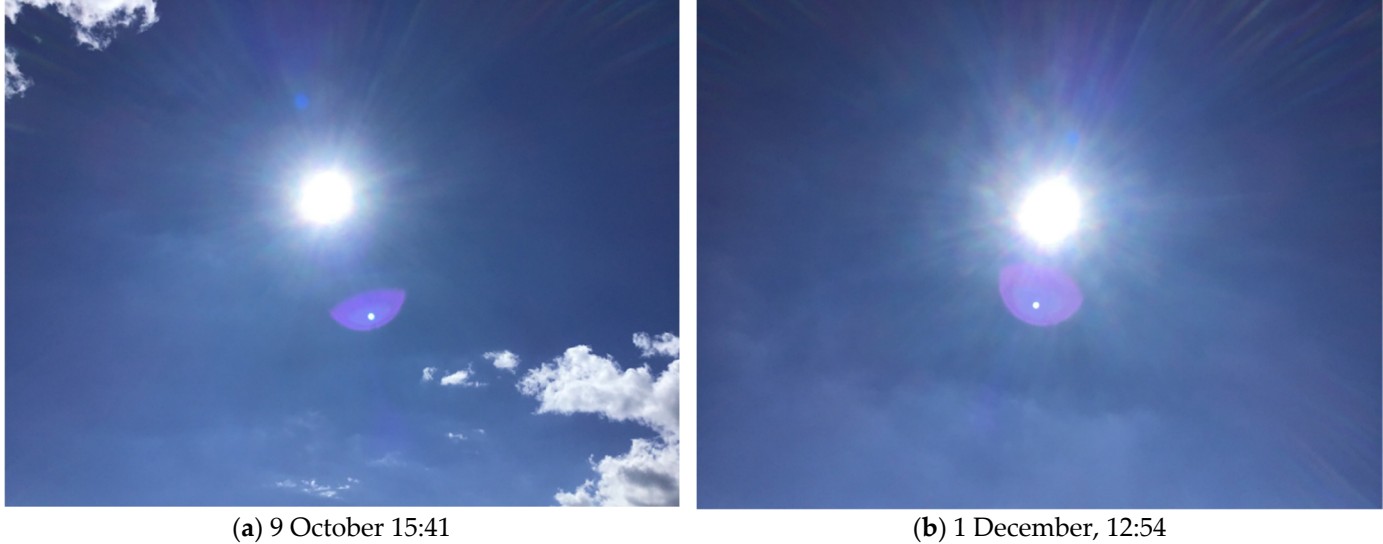

(**a**) 9 October 15:41      (**b**) 1 December, 12:54

**Figure 15.** Weather conditions for the two observations at the solar altitude angle of 41.263°.

In terms of the high-spatial resolution observations, which were taken on the afternoon of 3 December as an example, a number of broken clouds appeared in the sky at around 15:00 during the observation period, as shown in Figures 16 and 17 depicts the images (and their horizontal profiles) obtained during the observation at around 15:17, when the sky was nearly clear and when clouds were present. It can be observed that HARIS has a sensitive response when the cloud obscures the sun, which indicates that HARIS is capable of high-spatial resolution remote sensing observations of the sky.

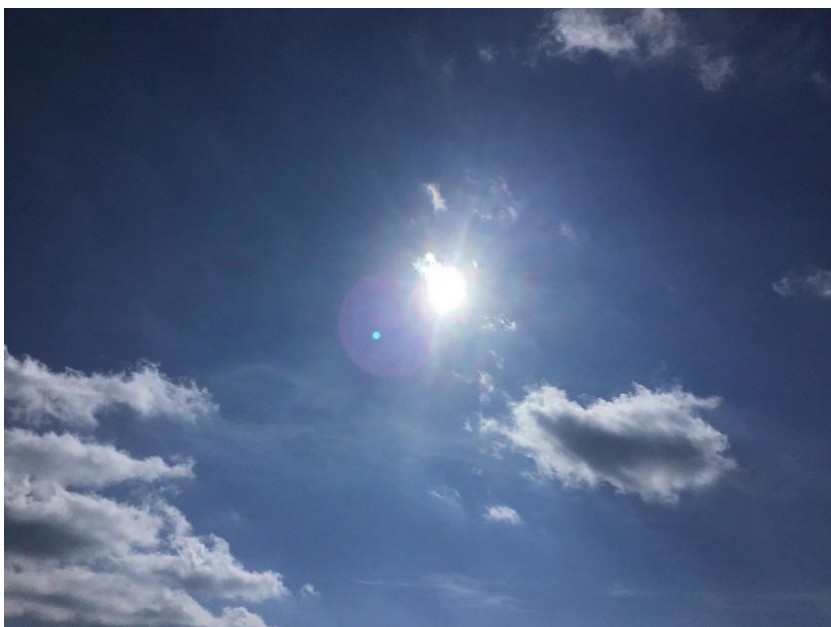

**Figure 16.** Weather at the time of the observation of 3 December at 15:00.

The continuous observation capability of HARIS was also verified during the field tests. Figure 18 shows the radiance ratios of the absorbed trough at 761 nm to the unabsorbed band at 758 nm that was obtained from the observations in the relatively clear weather conditions of the afternoons of 2 December and 3 December; the weather conditions at some moments during these observations are shown in Figure 19. From the observed data, it is clear that, as the solar altitude angle decreases, the solar radiation needs to pass through a longer path in the atmosphere before it is received by HARIS. Consequently, the absorption in the oxygen A-band was more intense at low solar altitude angles, which is reflected in the spectral data, since the ratio of the radiance of the 761 nm band to that of the 758 nm band decreases with descending solar altitude angles. The comparison of the observations for those two days also shows that a similar effect arises during the presence of clouds in the observation path.

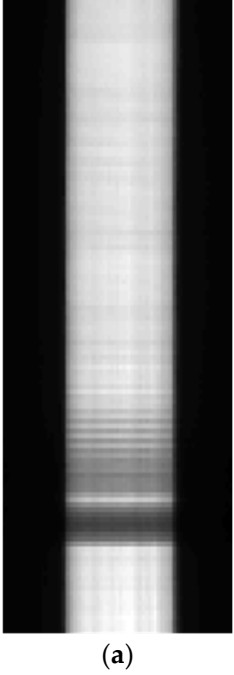
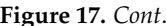
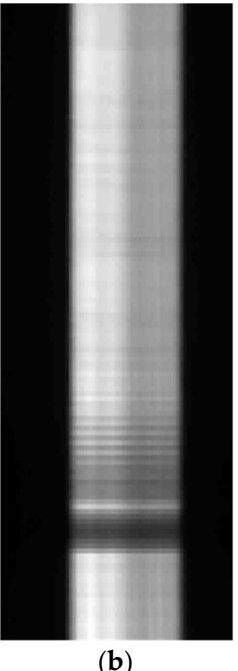

(**a**)                    (**b**)

**Figure 17.** *Cont.*

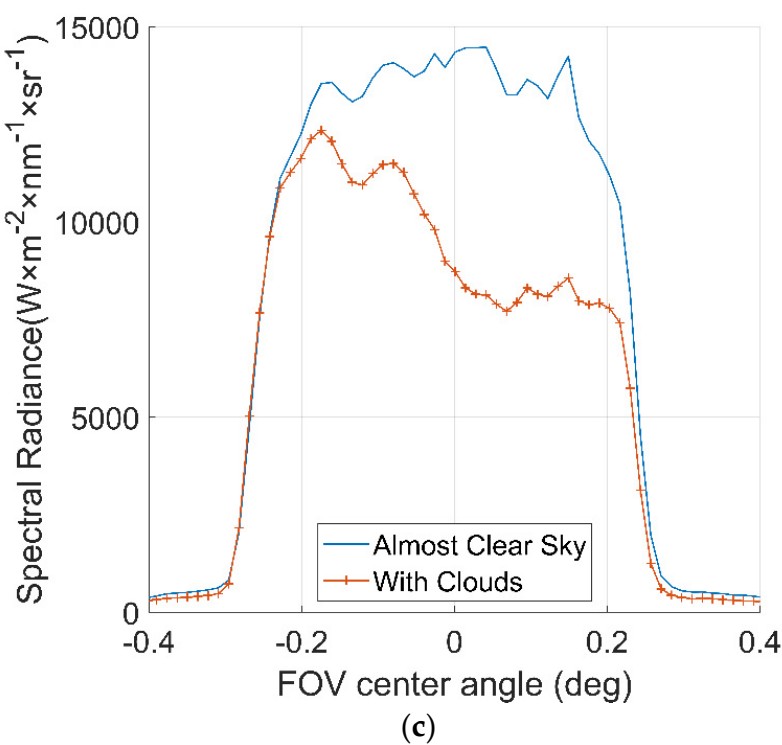

**Figure 17.** Image when (**a**) the sky is almost clear, (**b**) the clouds appear, and (**c**) the horizontal profile, which near to 758.4 nm.

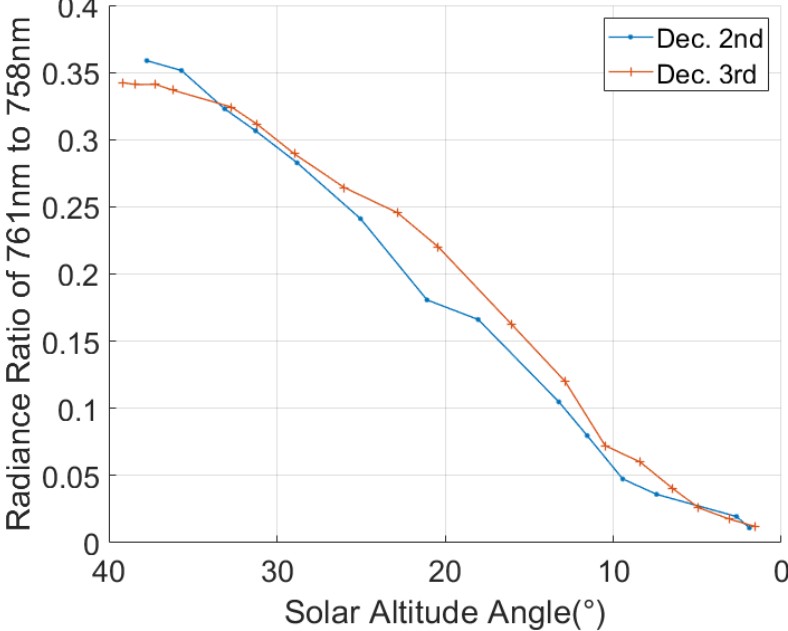

**Figure 18.** Radiance data for the different solar altitude angles that are obtained from consecutive observations on the afternoon of 2 and 3 December.

The observed data also display a high SNR. Figure 20 provides the SNR data that were calculated from 100 consecutive frames of data, during the observation at 15:41 on 9 October. For this period, the data acquisition rate was set at 43 Hz, the acquisition time of the 100 frames was less than 3 s, and the sky radiation signal was presumed to be basically unchanged during this period (and thus had little influence on the SNR calculations). Near the unabsorbed region around 758 nm, the instrument SNR can be more than 500. With a change in the atmospheric absorption and the instrument response, the lowest SNR appears

at the bottom of the absorption trough close to 761 nm, in which the minimum SNR is 283.8, and the average SNR of the full imaging area is 438.93.

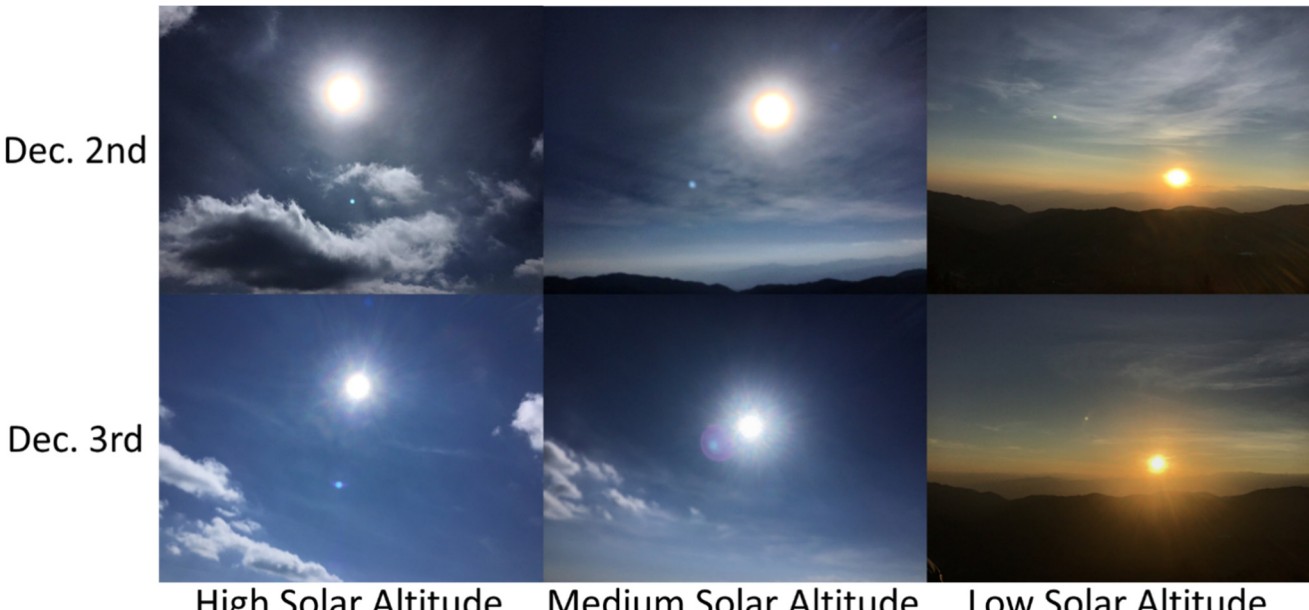

**Figure 19.** Weather during the continuous observations of the afternoon of 2 and 3 December.

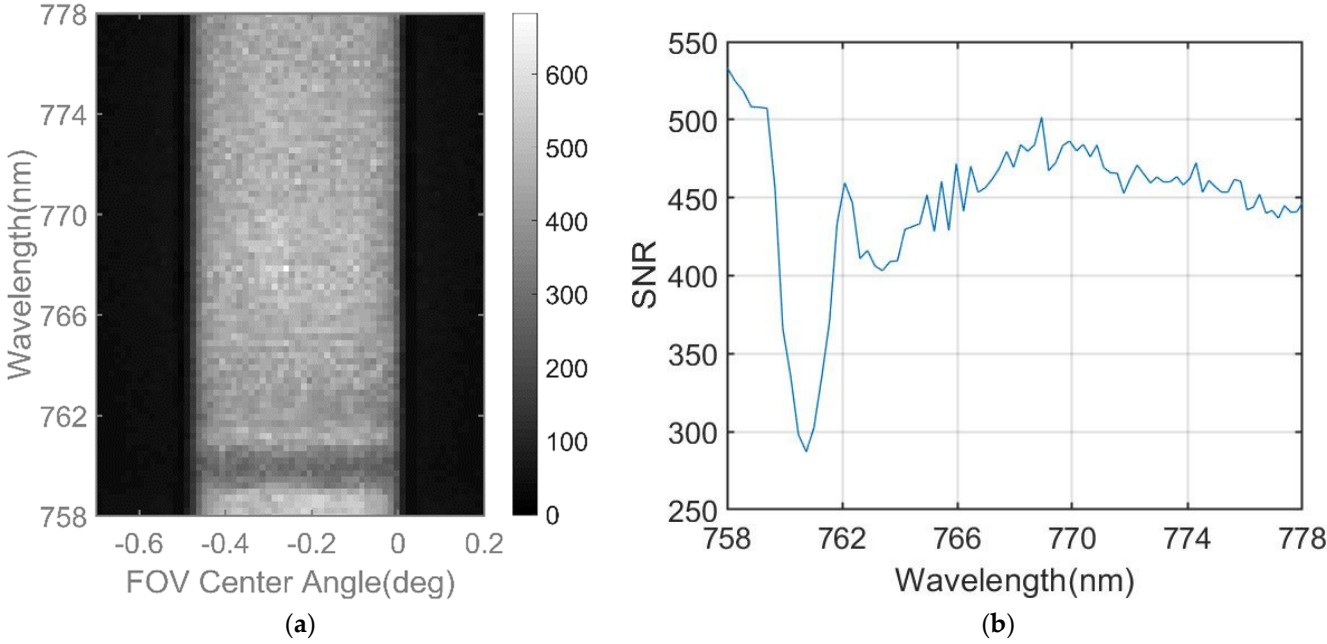

**Figure 20.** SNR of the 100 frames of data collected at 15:41 on 9 October (**a**) and the mean SNR of the full imaging area (**b**).

## 4. Discussion

The comparison between HARIS and the other oxygen A-band observation spectrometers mentioned in the references is presented in Table 1. The compared parameters include the size of the instrument, observation band, spectral resolution, maximum SNR, cooling condition and imaging condition. The table show that HARIS covers the oxygen A-band with acceptable spectral resolution and obtains a high SNR in a compact size. This ensures that the instrument can obtain high-quality observations with a maximum SNR > 500, an average SNR > 400, and a minimum SNR > 250.

**Table 1.** Performance comparison of oxygen A-band spectrophotometers.

| Instrument | Reference | Instrument Dimensions (Mm) | Observation Bands (nm) | Spectral Resolution (nm) | Maximum SNR | Cooling Condition | Imaging Condition |
|---|---|---|---|---|---|---|---|
| HARIS | This work | 230(L) × 270(W) × 170(H) | 758–778 | 0.33 | >500 | without CMOS cooling | Direct imaging |
| HABS | Min et al. [23] | Specific dimensional parameters not mentioned | 759–769 | 0.016 | >1300 | cooling CCD to −70 °C | Fiber optic homogenization imaging |
| RSS | Min et al. [24] | not mentioned | 750–780 | 2.3 | not mentioned | not mentioned | not mentioned |
| SPHABS | Xia et al. [26] | not mentioned | 759–770 | 0.016 | not mentioned | not mentioned | Fiber optic homogenization imaging |
| DGSS | Li et al. [27] | 1050(L) × 480(W) × 350(H) | 758–778 * | 0.07 * | >300 | cooling CMOS to −10 °C | Fiber optic homogenization imaging |

\* DGSS has two different observation bands and resolutions, and the parameters marked here are those of the instrument dedicated to the oxygen A-band observation.

Despite the trade-offs in optical parameters made in the miniaturization of HARIS, the slightly lower spectral resolution of this instrument can still meet the needs of oxygen A-band observations. At the same time, the instrument improves its SNR by operating in the direct-to-sun observation mode, which greatly improves the data quality and compensates for the performance loss caused by the uncooling and miniaturization.

In addition, HARIS also adopts a direct imaging observation method that increases the spatial sampling interval to 0.0138°, allowing more detail, such as the status of clouds in the target area to be observed.

## 5. Conclusions

This paper initially introduces a compact optical system based on HARIS. Next, the process of wavelength calibrating of HARIS using a system that is composed of a supercontinuum laser and a monochromator was described; the instrument spectral measurement performance parameters were also given. Subsequently, the method of radiometric calibration of HARIS using an integrating sphere and a spectrophotometer was described; the results of the calibration were also given. Then, the method and the results of using a geometric calibration to determine and correct the image distortion were presented, which ensure that HARIS is capable of high-spatial resolution remote sensing observations. Finally, the field tests confirmed that HARIS is able to obtain oxygen A-band remote sensing data, with both a high spectral resolution (0.33 nm) and a high SNR (283.8). This system can thus be used for the efficient ground acquisition of oxygen A-band remote sensing data.

**Author Contributions:** Conceptualization, J.W., N.H. and H.C.; formal analysis, H.W., D.C., Z.Z., Y.S. and B.L.; investigation, H.W. and D.C.; methodology, P.W., G.L., D.C., Z.Z., Y.S. and B.L.; software, H.W., P.W. and Y.S.; supervision, G.L. and B.L.; validation, G.L. and B.L.; writing—original draft, H.W. and D.C.; writing—review & editing, H.W., Z.Z. and B.L. All authors have read and agreed to the published version of the manuscript.

**Funding:** This research was funded by the National Key R&D Program of China, grant number 2018YFB0504600, 2018YFB0504603 and the Strategic Priority Research Program of Chinese Academy of Sciences, grant number XDA 28050102.

**Institutional Review Board Statement:** Not applicable.

**Informed Consent Statement:** Not applicable.

**Data Availability Statement:** Not applicable.

**Acknowledgments:** The authors are thankful to Xiaoxu Wang for assistance with the language, Jifeng Li, Limin He and Zhanfeng Li for helping on the instrument tests, and every member of the HARIS team; all their contributions were necessary for completion of this paper.

**Conflicts of Interest:** The authors declare no conflict of interest.

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
