# Peer review of "A Transmissive Imaging Spectrometer for Ground-Based Oxygen A-Band Radiance Observation"

_photonics, doi:10.3390/photonics9100729_

Round 1
Reviewer 1 Report
The authors present a compact spectral imager that measures the O2 A-band spectrum through the open atmosphere, a technique that has important applications in environmental research.
line 127: "So, a total of 2040 (columns) and 550 (rows)..." - need to be replaced
Lines 363-294: Figure 17c – The 'x -axis' corresponds to 'FOV center angle (deg)' and not to ‘wavelength (nm)‘ as shown in the fig.
Lines 317-318 - "Near the unabsorbed region around 758 nm, the instrument SNR can be more than 500" - Would be nice to see the Allen deviation of this instrument (your choice)
Line 334: with both a high "spatial" (should be spectral) resolution (0.25 nm - "this is spectral sampling interval") your resolution is FWHM of 0.33/0.36 nm.
I suggest to make improvements to your "Conclusion":
1. It would have been helpful if you had presented a comparison between your instrument and the instruments in other papers you cite, such as "Ref 10, Ref 23,.. for example", in terms of SNR, spectral resolution, and A-band spectral coverage.
2. According to you, the novelty in your system is its compactness. How compact is it? When compared with other systems, what is the compromise in terms of SNR?
In my opinion, this last effort will maximize the impact of your paper
Good luck
Reviewer 2 Report
Wu et al. present an interesting apparatus for measuring the oxygen A-band range from the ground which has comparable resolution and smaller size compared to other spectrometers with similar functions. I would like to recommend accepting this paper after minor revision. Here are some questions and suggestions that I have.
1. In section 2.2, it is stated that pixel binning can help reach high-resolution without cooling the detector.
a) In the ref. 28, it is mentioned that the frame binning can also improve the resolution, will the frame binning also work for this study?
b) Can you show some quantitative results on how your pixel binning help to improve the resolution like a noise level comparison between current binning size, smaller binning size, or no binning?
c) Can you show an estimation of the thermal effect based on your spectrum?
2. Figures 8. and 12. are the detector responses images but can you add more details information on the pictures like axis title etc. for the reader to easily understand?
3. Figure 17 (b) image when the clouds appear. I would expect the image to be darker when the sunlight is partially blocked by the cloud. Can you explain why it shows a left bright – right dark image?
4. Please further polish the language to make this article fluent to read for international readers.
Reviewer 3 Report
The authors presented a compact imaging spectrometer for the oxygen A-band observation. The instrument design and calibration were given, which ensured the high SNR and high spectral resolution. Furthermore, the field test results were described and analyzed. In the end, the authors claimed that such developed method can be used to efficiently capture the oxygen A-band remote sensing data. However, I have several questions and concerns on the presentation:
1. The authors mentioned that they wanted to develop a compact and easily deployable instrument, however, what about other potential candidates for the oxygen A-band acquisition? And is there any drawback for the proposed system. Without knowing all these, it's hard to judge the efforts and worth presented in this work.
2. In the field test, the authors measured the data on two individual dates, Oct. 9 and Dec. 1. I was wondering if there's any particular reason to choose these two days. And their solar altitude angles are the same or different? This 's confusing.
3. Is there any conclusion from the figure 14. I didn't see anything in the text.
4. I also think it could be worthwhile to have a comparison between other methods and this system on the oxygen A-band data detection. A table should suffice.
Overall, I would like to recommend the paper to be published after major revisions.
Round 2
Reviewer 3 Report
The concerns have been properly addressed in the revisions.